# Nonhost Resistance of *Thinopyrum ponticum* to *Puccinia graminis* f. sp. *tritici* and the Effects of the *Sr24*, *Sr25*, and *Sr26* Genes Introgressed to Wheat

Lyudmila Plotnikova * , Valeria Knaub and Violetta Pozherukova 

Agrotechnological Faculty, Omsk State Agrarian University Named after P.A. Stolypin, 644008 Omsk, Russia; vv.knaub06.06.01@omgau.org (V.K.); ve.pozherukova@omgau.org (V.P.)
* Correspondence: lya.plotnikova@omgau.org; Tel.: +7-(904)-5849083

**Abstract:** The damage to wheat crops by stem rust poses a threat to the food security of the world's population. The species *Thinopyrum ponticum* (Podpěra, 1902) (Z.-W. Liu and R.-C. Wang, 1993) is a nonhost for the stem rust fungus *Puccinia graminis* f. sp. *tritici* Erikss. and Henn. (*Pgt*). The *Sr24*, *Sr25*, and *Sr26* genes, transferred from the *Th. ponticum* to the wheat gene pool, protect cultivars from the disease in different regions of the world. The study of the nonhost resistance (NHR) of *Th. ponticum* and the effects of the introgressed *Sr24*, *Sr25*, and *Sr26* genes in wheat is important for breeding cultivars with durable resistance to stem rust. The aim of the research is to study the interaction of *Pgt* with *Th. ponticum* and common wheat lines with the *Sr24*, *Sr25*, and *Sr26* genes, in addition to determining the role of ROS in resistance. Wheat lines with *Sr24*, *Sr25*, and *Sr26* were resistant to the West Siberian *Pgt* population. Using cytological methods, it was found that the NHR of *Th. ponticum* and *Sr24*, *Sr25*, and *Sr26* led to inhibition of the most inoculum development on the plant surface. This was mainly due to the suppression of the appressoria development and their death at the stage of penetration into the stomata. Upon contact of *Pgt* appressoria with stomatal guard cells, the generation of the superoxide anion $O_2^{\bullet-}$ was revealed. This interaction is similar to the stomatal immunity of *Arabidopsis thaliana* to non-pathogenic bacteria. The results of our studies show that the *Sr24*, *Sr25*, and *Sr26* genes reproduce the action of some NHR mechanisms in wheat.

**Keywords:** *Thinopyrum ponticum*; nonhost; stem rust; wheat; introgression; *Sr* genes; PTI-ETI; PRRs; ROS generation; stomatal immunity; pre-invasion barriers

## 1. Introduction

Wheat is one of the three most important cereals that provide nutrition to the world's population. In 2019, wheat crops occupied an area of 216 million hectares in the world [1] and were cultivated on all continents (excluding Antarctica). To provide food for the growing population, it will be necessary to increase wheat grain production by 70% by 2050 [2]. Fungal diseases cause significant crop losses and a decrease in the quality of wheat bread products [3]. Potential damage from diseases is estimated at 18%, although it is reduced to 13% due to the use of fungicides [4].

Rapidly evolving rust fungi pose a constant threat to wheat crops. Wheat is damaged with leaf, stem, and stripe (yellow) rusts. Their causal agents are the biotrophic fungi *Puccinia triticina* Erikss., *P. graminis* f. sp. *tritici* Erikss. and Henn., and *P. striiformis* Westend f. sp. *tritici* Erikss., respectively. Leaf rust regularly affects wheat in all regions of the world and causes average losses up to 10%, and in outbreaks up to 60% of the crop [5]. The development of stem rust has been suppressed for several decades due to the large-scale cropping of cultivars protected by the *Sr31* gene transferred from the rye *Secale cereale* L., sometimes in combination with the adult plant resistance *Sr2* or other genes [6]. However, in 1998, the Ug99 race of *P. graminis* f. sp. *tritici* (*Pgt*) appeared in Uganda, overcoming the *Sr31* [7]. Over time, the Ug99 race and its derivatives (Ug99 group) spread to the countries

of Africa, Middle East and Europe [6,8]. With the epidemics of stem rust, grain losses reach 100%, so the disease is recognized as a threat to the food security of humanity [6,9]. The spread of stripe rust has long been limited to areas with a cool, humid climate. However, after the appearance of high-temperature resistant clones of *P. striiformis* f. sp. *tritici* in the 2000s, the harmfulness of stripe rust increased in many regions of the world [10]. In this regard, it is important to know about defense mechanisms that ensure the durable resistance of wheat to rust diseases.

In Russia, wheat stem rust outbreaks were noted in 2004 and 2006, and moderate development appeared in 2013 and 2014 in the Volga Region [11]. A strong rust epidemic was noted in 2016 on crops on the right bank of the Volga river [12,13]. The Ural region and Western Siberia are important grain producers, with spring wheat accounting for more than 40% of crops [14]. In the neighboring regions of the Republic of Kazakhstan, wheat occupies about 15 million hectares [15]. In Western Siberia, stem rust was not registered for a long period until 2008, and later it manifested locally. An increase in disease severity was noted at the end of summer in 2014 [16]. In 2015, an epidemic broke out on large-scale wheat crops in Western Siberia and Northern Kazakhstan, which led to the loss of 50% of the grain [17]. *Pgt* populations existing in Russian regions differ from African and European ones, and their evolution continues independently of the Ug99 race [8,18,19]. In Western Siberia, the isolated *Pgt* population is subdivided into two subpopulations (Omsk and Altai ones). The Omsk *Pgt* population is highly aggressive, and its pathotypes have a large number of virulence genes [19,20].

Growing resistant cultivars is the most economically efficient and environmentally friendly way to manage rust diseases [3,21,22]. However, cultivar resistance can be quickly overcome when new virulent races appear in fungal populations. The practice of crop production has shown that overcoming resistance has accelerated in agrocenoses. Important factors in the pathogen's evolution are large-scale monocropping, low genetic heterogeneity of cultivars, and intensive production technologies [23,24]. Increasing genetic diversity is important to protect wheat cultivars from rapidly evolving pathogens [25,26]. At first, cultivars were protected with wheat resistance genes, and since the 1960s, alien genes have been used [27]. Traditionally, relative cereals are used to enrich the wheat genetic pool [28–33]. The most difficult problem is transferring genetic material from species that differ from wheat (BBAADD) by one or more genomes (such as genera *Aegilops*, *Secale*, *Hordeum*, *Thinopyrum*, *Elymus*, and *Haynaldia*) [27]. In recent years, the influx of genetic material of the *Agropyron*, *Thinopyrum*, *Dasypyrum*, and *Pseudoroegneria* species into the wheat gene pool has increased. These species have different combinations of genomes: J(=E), St, W, Y, X, V, H, P [25,34–37].

*Thinopyrum* species have a perennial lifestyle and differ in their resistance to soil salinity, drought, and extreme temperatures [25,29,38–40]. The *Thinopyrum* genus is of most interest as a resource of resistance genes to a wide range of wheat diseases. The first work on the distant hybridization of common wheat with *Thinopyrum* species (=*Agropyron*) was carried out in Russia by N.V. Tsitsin in the 1920–1930s. The most promising species for wheat breeding were *Thinopyrum ponticum* (Podpěra, 1902) (Z.-W. Liu and R.-C.Wang, 1993) (=*Agropyron elongatum* Host ex *P. Beauvois*, 1812) and *Th. intermedium* (Host) Barkworth and D.R. Dewey [40]. Later, the same species were hybridized with wheat in the USA, Germany, Canada, and China [41,42]. Wheat lines with supplemented or substituted chromosomes of the *Thinopyrum* species and translocations of various sizes were created [43–45]. *Th. ponticum* differs from common wheat in its genome constitution ($2n = 10x = 70$, JJJJJJJ$^S$J$^S$ or EeEeEbEbExExStStStSt) [46]. It was used as a source of resistance genes to stem (*Sr24*, *Sr25*, *Sr26*, *Sr43*, and *Sr61*), leaf (*Lr19*, *Lr24*, and *Lr29*) and stripe (*Yr50*) rusts. Four of them are present in complex translocations—*Lr19/Sr25* and *Lr24/Sr24*. The *Lr38* and *Sr44* genes were transferred from *Th. intermedium* [30–32].

The *Thinopyrum* genes have been actively used in wheat breeding since the 1970s. By 2012, there were about 12.500 cultivars and lines the world, mainly protected by the *Th. ponticum* genes (93%). In the USA, about half of the cultivars were protected by the

translocation *Lr24/Sr24* and 12% the translocation *Lr19/Sr25* [47]. In Western Europe, tall wheatgrass genes are used less frequently, but 17% of Germanic cultivars have the *Lr24/Sr24* genes (sometimes combined with *Sr31* and *Sr38*) [48]. In Russia, the translocation *Lr19/Sr25* and the substituted chromosome 6Agi from *Th. intermedium* are mainly used in cultivars in the Volga region [49]. In Australia, *Sr26* is used in cultivars (82%), and less often the translocations *Lr24/Sr24* and *Lr19/Sr25* (20% and 6%, respectively). In South Africa and Egypt, about 5% of varieties have the *Lr24/Sr24* genes [47].

In connection with the regular overcoming of cultivar resistance, it is of interest to study the resistance of nonhost species (NHR) [50,51]. Pathogenic fungi infect a limited range of plant species that have become "hosts" for them, but the rest show immunity and remain "nonhosts", and their resistance is rarely overcome [51,52]. In this regard, the study of NHR regulation and mechanisms is of great theoretical and practical interest. In the 1970s, the NHR model based on the concepts of "basic compatibility" and "basic resistance" was proposed. It was proposed that, in the course of co-evolution, specialized pathogens gradually acquire genes and properties to be able to develop on plants, i.e., basic compatibility. After that pathogen overcomes a set of basic resistance mechanisms, the species becomes the "host" [53,54]. A significant part of the basic resistance consists of preformed constitutional barriers, which include the physical and chemical plant features [55]. The pathogen's inhibition in the early stages of development is typical for NHR [50,56]. After NHR is overcome, plants can be protected by host (cultivar) resistance mechanisms, in accordance with the "gene-for-gene" theory of H. Flor [57,58].

Later, based on molecular studies, a two-stage model of plant immunity was developed based on non-host and host (cultivar) immunity [59,60]. At first, plant PRRs (Pattern Recognition Receptors) localized on the cell surface recognize the metabolites of non-pathogenic microorganisms–MAMPs (Microbe-Associated Molecular Patterns), non-specialized pathogen's –PAMPs (Pathogen-Associated Molecular Patterns), and DAMPs (Damage-Associated Molecular Patterns). After that, signaling is activated, and PAMP-triggered immunity (PTI) is triggered. To overcome PTI, pathogens have to acquire a set of effectors to deactivate the resistance mechanisms. Host resistance is determined by NLR receptors localized in the cytoplasm. NLRs recognize race-specific effectors, as a result the second level of immunity—ETI (Effector-Triggered Immunity) is activated. ETI is usually accompanied by reactive oxygen species (ROS) generation and a hypersensitive reaction (HR) [61,62]. The earliest responses of plants after the recognition of PAMPs and effectors are $Ca^{2+}$ influx, ROS generation, and induction of mitogen-activated protein kinase (MAPK) cascades [61,63–65]. The PTI-ETI model has been well studied on *Arabidopsis thaliana*, but it needs to be tested in other pathosystems. In this regard, it is of interest to study the regulation and mechanisms of cereal NHR and resistance genes to rusts introgressed from them.

Rust fungi form a set of infection structures for interactions with plants [66]. After adhesion to the surface of susceptible plants, the urediniospores form growing tubes directed to the stomata and develop appressoria on them. After penetration into the stomata, pathogens form substomal vesicles, infection hyphae, haustorial mother cells and haustoria to absorb nutrients from the mesophyll cells [67]. Urediniopustules with the next generation of urediniospores are formed on susceptible plants after a few days. The development of rust fungi on non-hosts depends on the phylogenetic distance to the host species. *Puccinia* spp. on non-host cereals stop development before the haustorium penetration into plant cells (pre-haustorial resistance) [68,69]. *Uromyces* spp. on closely related *Fabaceae* spp. form microcolonies whose development is accompanied by HR [70]. Pre-haustorial resistance is considered to be a manifestation of an extreme incompatibility of rust fungi with nonhosts [50,56].

Despite the active use of the genetic material of *Th. ponticum* to protect wheat, the regulation and mechanisms of NHR to stem rust and the effects of introgressed genes have not been studied. In this regard, the aim of this research is to study the interaction

of *Pgt* with *Th. ponticum*, and common wheat with the *Sr24*, *Sr25*, and *Sr26* genes, and to determine the role of ROS in plant resistance.

## 2. Materials and Methods

### 2.1. Plant Material

The accessions of *Th. ponticum*, interspecific hybrid, and wheat–wheatgrass hybrids (WWHs), and cultivars and lines of spring common wheat *Triticum aestivum* L. were used in the study. The accessions of *Th. ponticum* originating from Russia and South Africa were received from the Collection of the Main Botanical Garden of the Russian Academy of Sciences (Moscow, Russia). The interspecific hybrid ((*Triticum durum* × *Th. ponticum*) × *T. aestivum* cv. Pyrotrix 28) was originated on the basis of the Russian sample, at Omsk State Agrarian University (Omsk SAU, Omsk, Russia). This hybrid was crossed with susceptible cultivars of common wheat, and a set of wheat–wheatgrass hybrids (WWHs) was selected [71]. Perennial samples of *Th. ponticum*, an interspecific hybrid, and WWHs were grown in the collection of Omsk SAU.

The effects of the genes introgressed from *Th. ponticum* were studied on the cultivars and near-isogenic lines (NILs) of spring common wheat—NIL-LMPG-*Sr24*, NIL-Tc*Lr24*/*Sr24* (RL6064), NIL-LMPG-*Sr25* (CI 14048), NIL-Tc*Lr19*/*Sr25* (RL6040), Eagle-*Sr26* (McIntosh), and NIL-LMPG-*Sr26*. Spring common wheat Pamyati Azieva and Chernyava 13 were used as indicators of susceptibility to stem rust in the field and laboratory experiments.

### 2.2. Estimation of Stem Rust Development in Field and Laboratory Conditions

Field experiments were carried out in the southern forest steppe of Western Siberia (Omsk, 54.58_N, 73.24_E) in 2015–2022. The samples of spring common wheat were sown in the 3rd ten days of May on plots of 1 m$^2$, with a seeding density of 500 grains/m$^2$. The perennial accessions of *Th. ponticum*, the interspecific hybrid, and WWHs were sown in 2010.

The stem rust development was evaluated on adult plants (at Zadoks scale, ph. 60–82) in field conditions with a natural infection background. The estimations were carried out in dynamics with a 10-day interval, starting with the first symptoms of rust and finishing at the wax ripeness stage. In 2022, a field assessment was not carried out, as, due to a severe drought, rust appeared after the ripening of wheat. The assessments of stem rust were carried out according to the CIMMYT methodology. Infection type (IT) was estimated according to modified Stakman's scale and disease severity (SR) on a 0–100% modified Peterson's scale [72,73].

Laboratory experiments were carried out using the Omsk *Pgt* population. Infected wheat stems were collected in 2020 and 2022 at the experimental fields of the Omsk SAU. *Pgt* urediniospores were reanimated and propagated on the susceptible cv. Chernyava 13. The accessions of *Th. ponticum*, the interspecific hybrid, and WWHs were infected with the propagated inoculum of 2020 and 2022 *Pgt* populations. Wheat cultivars were infected with monopustule isolates obtained from the same *Pgt* populations.

Seedling tests were carried out on 10-day-old plants grown in pots. Inoculation was made by spraying with a suspension of urediniospores (10–12 thousand spores/mL) in a 0.01% water solution of detergent Twin-80 (Sigma-Aldrich, Inc., St. Louis, MO, USA). The phenotyping of isolates from 2022 population was carried out using an International North American set of test lines for the differentiation of wheat stem rust races (*Pgt* differential set), in accordance with the standard methods [19]. Infected plants were incubated for 24 h at a temperature of 26–27 °C and then at 23–25 °C under illumination with a 16 h period, with an intensity of 10,000 lux until the appearance of pustules. ITs were determined by a modified Stackman's scale [74]: 0—without symptoms; ;—small necrotic spots; 1—medium necrotic spots and small pustules surrounded by medium necrotic zones; 1—small necrotic spots and small pustules with small necrotic zones; 2- —chlorotic spots and microscopic pustules surrounded by large chlorotic zones; 2—medium pustules surrounded by chlorotic zones; 2+—medium pustules; 3+—large pustules surrounded by chlorotic zone; 4—large

pustules. Plants with Its 0–2+ were considered resistant, and those with ITs 3–4 were considered susceptible.

The isolates belonged to the races TKRPF (avirulence to *Sr9b*, *11*, *24*, *30*, and *31*) and RFPTF (avirulence to *Sr6*, *9b*, *9e*, *11*, *24*, *30*, and *31*) (10 and 4 isolates, accordingly). Isolates of TKRPF (Nos. 2, 4, and 5) and RFPTF (Nos. 1 and 3) were used for cytological studies. The susceptible cv. Chernyava 13 (IT 4) was used as the control in laboratory studies.

*2.3. Cytological Methods*

Infected leaves were used for the cytological studies. The material was fixed by boiling in the lactophenol solution (lactic acid:phenol:glycerin:water:96% ethanol = 1:1:1:1:8) at 1, 2, 3, 5, and 10 days after inoculation. To detect the infection structures and HR, the leaves were stained with 1% aniline blue in lactophenol at a temperature of 60° C for 0.5 h. Then the staining was differentiated in a saturated chloral hydrate solution (2.5 g chloral hydrate/ml $H_2O$) at a temperature of 60 °C for 2–3 h [75]. Intact fungal structures turned dodger blue, damaged ones were dark blue, alive plant cells were light blue, and dead cells after HR were dark blue. The ROS effect on pathogenesis was studied on plants treated with salicylic acid (SA, Sigma-Aldrich, Inc., St. Louis, MO, USA,) or verapamil (inhibitor of $Ca^{2+}$ channel (Calbiochem Immunochemicals, Madison, WI, USA,). To induce the ROS generation, the leaves were sprayed with a 0.01% SA water solution (equal to 0.63 mM, 1 ml/10 leaves). To suppress $Ca^{2+}$ influx and ROS generation, a 0.01% water solution of verapamil (1 ml/ 10 leaves) was applied 12 h before inoculation [76]. The infected leaves of untreated plants were used as the control. ROS generation was determined by vital leaf staining with 0.1% nitroblue tetrazolium (NBT) (Acros Organics, Thermo Fisher Scientific, Branchburg, NJ, USA) or 0.02% 3,3′-diamino-benzidine tetrachloride (DAB) (Sigma-Aldrich, Inc., St. Louis, MO, USA). NBT formed an insoluble blue formazan in the presence of the superoxide anion $O_2^{\bullet-}$ or dehydrogenase activity, and DAB formed a cherry one in the presence of hydrogen peroxide $H_2O_2$. Dye solutions in dechlorinated water were vacuum-infiltrated into the leaves. After incubation for 30 min, the material was fixed in lactophenol [76].

The cytological studies were carried out using a Micmed-5 light microscope (LOMO, St.-Petersburg, Russia). To study the interaction of *Pgt* with the samples, five plants per variant were used. The results of the *Pgt* interaction with each plant were considered as a repetition of the variant. At all observation periods, the results of the development of 20–30 spores per plant were determined. To assess the effect of resistant plants on the *Pgt* development, the following indicators were determined (number): germinated/non-germinated spores, growing tubes with appressoria, appressoria on stomata and on the surface, substomal vesicles (SVs), and haustoria per colony. Colony sizes (length and width, μm) were measured in five days after inoculation. The colony square (μm$^2$) was calculated using the ellipse area formula ($S = \pi \cdot a \cdot b/4$, where $a$ and $b$ are the major and minor axis, respectively). The mean values of cytological data and the error of the mean ($M \pm SEM$) were calculated.

**3. Results**

*3.1. Estimation of Stem Rust Development in the Field Conditions and Laboratory*

The development of stem rust on the accessions of *Th. ponticum* and wheat with the *Sr24*, *Sr25*, and *Sr26* genes was estimated under changing phytopathological situations in 2015–2022. In the Omsk region, stem rust appears on wheat crops in late July–early August and reaches its maximum in the end of August. The enhancement of stem severity is noted at the wax ripeness stage. Weather conditions contributed to the intensive disease development in 2015, 2016, 2018, and 2019 (damage of susceptible cultivars 80–100S). In the arid 2017, 2020, and 2021 conditions, the severity reached 40–60S to the end of plant vegetation (Table 1). All samples of *Th. ponticum*, the interspecific hybrid, and WWHs were immune (R) during the epidemic of 2015 and in the following years (Table 1). In 2015, wheat lines with the *Sr24*, *Sr25*, and *Sr26* genes were affected to a weak or moderate degree at the first assessment (5–20 MR-MS), and the final assessment showed moderate

susceptibility (30–50 MS-S). In the following years, the decrease in the severity and ITs of the introgressive lines were noted. In 2017–2020, at the first assessment, wheat samples showed immunity or a weak lesion with resistant ITs (R, 5MR-M), and an increase in the severity up to 10–20% was determined only at the final stage. In 2021, weak rust development was noted only on LMPG-*Sr25* at the wax ripeness phase (5MR). The effect of the *Th. ponticum*'s introgressed genes was evaluated in different genetic backgrounds. The estimations showed that the effects of the *Sr24*, *Sr25*, and *Sr26* genes differed quantitatively in samples, but the Tc*Lr24*/*Sr24*, Tc*Lr19*/*Sr25*, and LMPG-*Sr26* lines were more resistant than others. These lines were used for further experiments.

**Table 1.** Results of the estimation of stem rust development on *Th. ponticum,* wheat–wheatgrass hybrids, and wheat samples with introgressed genes in the field (Western Siberia, Omsk, Russia) and in the laboratory.

| Sample | Field, Severity, and IT | | | | | | | | | | | | | Seedling Test, IT * | |
| --- | --- | --- | --- | --- | --- | --- | --- | --- | --- | --- | --- | --- | --- | --- | --- |
| | 2015 | | 2016 | | 2017 | | 2018 | | 2019 | | 2020 | | 2021 | | |
| | 15.08 | 25.08 | 05.08 | 15.08 | 15.08 | 25.08 | 15.08 | 25.08 | 14.08 | 25.08 | 15.08 | 25.08 | 25.08 | 2020 | 2022 |
| Pamyati Azieva | 60S | 100S | 30S | 80S | 30S | 50S | 60S | 100S | 50S | 100S | 20S | 50S | 20S | 4 | 4 |
| Chernyava 13 | 60S | 100S | 40S | 80S | 40S | 60S | 60S | 100S | 60S | 100S | 20S | 50S | 40S | 4 | 4 |
| *Th. ponticum* Russia, Africa | R | R | R | R | R | R | R | R | R | R | R | R | R | 0 | 0 |
| IH ** | R | R | R | R | R | R | R | R | R | R | R | R | R | ; | 0 |
| WWH-1 *** | R | R | R | R | R | R | R | R | R | R | R | R | R | ; | 0 |
| WWH-2 | R | R | R | R | R | R | R | R | R | R | R | R | R | 0 | ; |
| WWH-3 | R | R | R | R | R | R | R | R | R | R | R | R | R | ; | ; |
| WWH-4 | R | R | R | R | R | R | R | R | R | R | R | R | R | 0 | 0 |
| WWH-5 | R | R | R | R | R | R | R | R | R | R | R | R | R | ; | ; |
| WWH-6 | R | R | R | R | R | R | R | R | R | R | R | R | R | 0 | 0 |
| LMPG-*Sr24* | 20MS | 50MS | 10MR | 30MS | 5MR | 30MS | 5MR | 30M | R | 30MS | 5MR | 10S | R | ;, ;1, 2 | ;1, 2- |
| Tc*Lr24*/*Sr24* | 10MS | 40S | 10MR | 30MS | R | 10MS | R | 10MR | R | 20M | R | 10MR | R | ;, ;1 | ;1, 2- |
| LMPG-*Sr25* | 20MS | 50S | 10MR | 20MR | 5M | 10M | 5M | 10M | 5MR | 20M | 5MR | 20M | 5MR | ;1, 2- | ;, ;1 |
| Tc*Lr19*/*Sr25* | 10MS | 30MS | R | 5MR | R | 5M | R | 5M | 5MR | 10M | R | 10M | R | ;1, 2- | ;1, 2- |
| Eagle-*Sr26* | 5MR | 40S | 5MR | 30MS | R | 10MR | R | 10M | R | 10M | R | 5MS | R | ;1, 2- | ;1, 2- |
| LMPG-*Sr26* | 10MS | 40S | 10M | 50MS | R | 5MR | R | 5M | R | 5M | R | 10MR | R | ;1, 2- | ;1, 2- |

Note: field conditions, natural background (Omsk, Russia); * Omsk *Pgt* population. ** IH—interspecific hybrid ((*Triticum durum* × *Th. ponticum*) × *T. aestivum* cv. Pyrotrix 28); *** WWH—wheat–wheatgrass hybrid; IT—infection type; 0—without symptoms; ;—small necrotic spots; ;1—medium necrotic spots and small pustules surrounded by medium necrotic zones; 2- —chlorotic spots and microscopic pustules surrounded by large chlorotic zones; 2—medium pustules surrounded by chlorotic zones; 4—large pustules.

In seedling tests in 2020 and 2022, the accessions of *Th. ponticum*, the interspecific hybrid, and WWHs were immune to disease. Their leaves showed no signs of lesions or small necrotic spots without pustules (0 and ;, respectively). Resistant ITs were noted on all wheat lines with the *Sr24*, *Sr25*, and *Sr26* genes. In response to inoculation with urediniospores, necrotic spots or small pustules surrounded by necrotic or chlorotic zones (ITs ;, ;1, 2-) appeared (Table 1 and Figure 1).

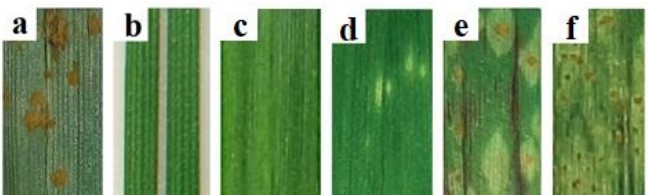

**Figure 1.** Development of stem rust on seedlings of the cv. Chernyava 13 (**a**); *Th. ponticum* (**b**); WWH-6 (**c**); WWH-2 (**d**); line LMPG-*Sr24* (**e**); and LMPG-*Sr25* (**f**).

The ITs of the Tc*Lr24*/*Sr24* and Tc*Lr19*/*Sr25* and LMPG-*Sr26* lines were further studied in seedling tests when infected with randomly selected *Pgt* isolates from the 2020 and 2022

populations (13 and 14 isolates, respectively). In 2020, most of the isolates were unable to form pustules (immunity) on lines with the *Sr24*, *Sr25*, and *Sr26* genes (ITs 0, ;) (90%, 61%, and 61% of isolates, respectively) (Figure 2a). The remaining isolates showed resistant ITs (;1, 1, 2-) with necrotic spots or single small pustules surrounded by necrotic or chlorotic zones of various sizes. In 2022, a smaller proportion of isolates with immune ITs was noted (26–35%), but more isolates formed small pustules with necrosis or chlorosis (ITs ;1, 1). At the same time, on the lines with *Sr24* and *Sr25*, more isolates caused IT with a chlorotic reaction (ITs 2-), and on the *Sr26* with a necrotic one (IT ;1).

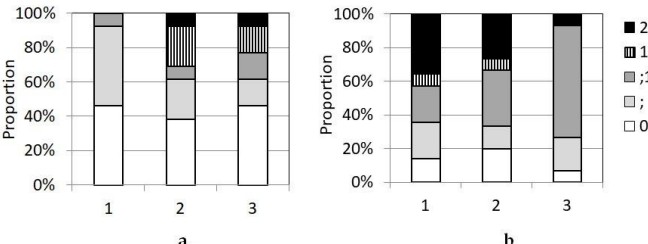

**Figure 2.** Distribution by infection types of common wheat introgressive lines infected with the *P. graminis* f. sp. *tritici* isolates in 2020 (**a**) and 2022 (**b**). 1—Tc*Lr24*/*Sr24*; 2—Tc*Lr19*/*Sr25*; 3—LMPG-*Sr26*. IT: 0—without symptoms; ;—small necrotic spots; ;1—medium necrotic spots and small pustules surrounded by medium necrotic zones; 1—small necrotic spots and small pustules with small necrotic zones; 2- —chlorotic spots and microscopic pustules surrounded by large chlorotic zones.

### 3.2. Interactions between P. graminis f. sp. tritici and Th. ponticum, and WWHs

To understand the mechanisms of NHR, the development of fungal infection structures and plant reactions were studied. On the surface of susceptible wheat cv. Chernyava 13, the main part of the spores (93%) germinated and formed growing tubes (Figure 3a and Table 2). Most of the growing tubes (87%) formed appressoria to penetrate into the stomata. About 90% of the appressoria were located on the stomata, which indicates the successful orientation of the growing tubes on the leaf surface. Most appressoria (85%) relocated their cytoplasm into the substomal vesicles (SVs), and empty shells remained on the guard cells (Figure 3a). SVs formed branched infection hyphae with specialized haustorial mother cells (HMC) at the tips. HMCs formed haustoria in the mesophyll cells (Figure 3b).

The NHR mechanisms were studied using the cases of *Pgt* interaction with two accessions of *Th. ponticum*, the interspecific hybrid, and the WWHs. The leaves of *Th. ponticum* and *T. aestivum* differed significantly in morphology. The narrow leaves of the tall wheatgrass had a relief surface due to the dense arrangement of vascular bundles (Figure 1b). The spore germination on the two accessions of *Th. ponticum* and hybrids did not significantly differ from the control (Table 2). However, the appressoria formation on the *Th. ponticum* was suppressed 3.2–6.3 times compared to that on wheat, and the effect was more pronounced in the Russian accession. In addition, most of the appressoria were located on the surface (about 60%), but not on the stomata (Table 2, Figure 3c). This indicates a violation of the growing tube's orientation to the stomata of *Th. ponticum*. On the Russian accession of the tall wheatgrass, development finished at the stage of appressoria penetration or without attempts of invasion into the stomata (Figure 3d,e). On the African accession, the single appressoria (7%) penetrated into the stomata and stopped at the SV stage (Figure 3f), and the rest died on the guard cells. Two days after inoculation, the cytoplasm of most appressoria and SVs was intensively stained, which indicates the destruction of the fungus membranes and cytoplasm (Figure 3d). On the interspecific hybrid, the intensity of appressoria formation was at the level of the African accession of *Th. ponticum* (26%), but the orientation of growing tubes and penetration into the stomata were improved (up to 70%). When infecting three WWH-3, WWH-4, and WWH-5, the intensity of appressoria formation increased compared to the parent accession of *Th. ponticum* but was lower than in the control (by 2.6–2.8 times). On the rest WWHs, the appressoria formation was more successful, but still weaker than on wheat (by 1.3–1.5 times). Fungal

growing tubes oriented to the stomata of WWHs more successfully (60–70%), but on all hybrids, the development finished at SVs. The intensity of penetration into the stomata and SV formation ranged from 0 to 13.5%. Thus, the incompatibility of *Pgt* with *Th. ponticum* and distant hybrids was manifested at three stages of the interaction: orientation of the growing tubes to the stomata, the development of appressoria, and penetration into the stomata. Therefore, the incompatibility of interaction led to the fact that only 0.7–3.1% of the inoculum applied to the leaves penetrated into the stomata, and the rest died on the leaf surface (Table 2).

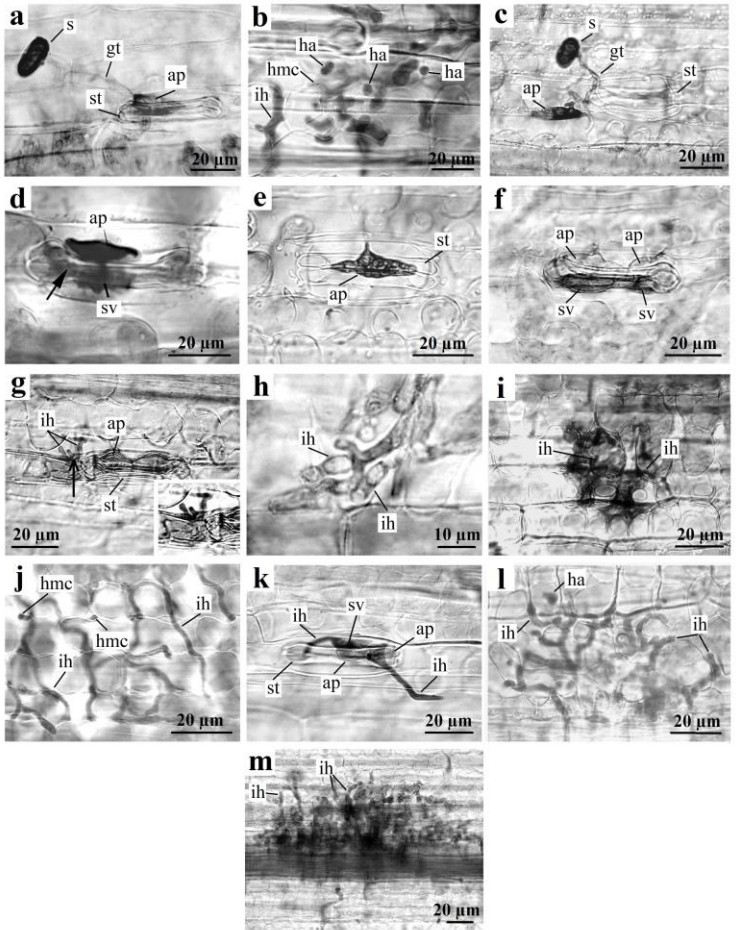

**Figure 3.** Development of the infection structures of *P. graminis* f. sp. *tritici* on wheat cv. Chernyava 13 (**a**,**b**,**l**), *Th. ponticum* (**c**–**f**,**k**), and introgressive lines (**g**–**j**,**m**). (**a**)—germinated spore, growing tube, and empty appressorium on the stoma; (**b**)—colony with branched hyphae, haustorial mother cells, and haustoria in the cells; (**c**)—dead appressorium on the leaf surface next to the stoma; (**d**)—appressorium and substomal vesicle with destroyed cytoplasm on the stoma; (**e**)—dead appressorium on the stoma; (**f**)—two empty appressoria and substomal vesicles with cytoplasm; (**g**)—microcolony below the stoma of the line with the *Sr24* (arrow; in the frame); (**h**)—vacuolized fungal hyphae in the tissue of the line with the *Sr25*; (**i**)—changed plant cells in the zone of small died colony; (**j**)—hyphae without haustoria in the tissue of the line with the *Sr26*; (**k**)—two appressoria with two infection hyphae in *Th. ponticum* treated with verapamil; (**l**)—colony with suppressed growth in the tissue of the cv. Chernyava 13 treated with verapamil; (**m**)—colony in the tissue of the line with the *Sr25* treated with SA. Aniline blue staining.

**Table 2.** Results of the development of the infection structures of *P. graminis* f. sp. *tritici* on the leaves of the *Th. ponticum* and distant hybrids.

| Sample | IT | Proportion of Germinated Spores, % | Proportion of Growing Tubes with Appressoria, % | Proportion of Appressoria, % | | Proportion of Inoculum Penetrated into Stomata, % |
|---|---|---|---|---|---|---|
| | | | | On Stomata from the Total Number | Germinated from the Number on the Stomata | |
| Chernyava 13—control | 4 | 91.0 ± 3.6 | 87.0 ± 4.5 | 89.0 ± 3.4 | 85.0 ± 0.9 | 59.9 |
| *Th. ponticum*, Russia | 0 | 87.1 ± 4.4 | 14.1 ± 0.2 | 39.3 ± 3.5 | 0.0 | 0.0 |
| *Th. ponticum*, Africa | ; | 88.4 ± 4.4 | 27.2 ± 1.4 | 40.0 ± 4.0 | 7.1 ± 4.6 | 0.7 |
| IH | 0 | 82.6 ± 4.1 | 26.0 ± 1.3 | 58.6 ± 2.9 | 13.5 ± 4.3 | 1.7 |
| WWH-1 | 0 | 77.9 ± 3.9 | 55.9 ± 2.8 | 60.2 ± 3.0 | 11.9 ± 4.4 | 3.1 |
| WWH-2 | ; | 76.8 ± 3.8 | 42.0 ± 2.1 | 63.6 ± 3.2 | 9.2 ± 2.5 | 1.9 |
| WWH-3 | ; | 79.1 ± 3.9 | 32.0 ± 1.6 | 70.3 ± 3.5 | 5.4 ± 4.7 | 1.0 |
| WWH-4 | 0 | 85.3 ± 4.3 | 34.0 ± 1.7 | 63.1 ± 3.2 | 11.0 ± 4.5 | 2.3 |
| WWH-5 | ; | 83.8 ± 4.2 | 31.7 ± 1.6 | 67.3 ± 3.4 | 10.8 ± 4.5 | 1.9 |
| WWH-6 | 0 | 78.8 ± 3.9 | 46.2 ± 2.3 | 70.5 ± 3.5 | 0.0 | 0.0 |

Note: IH—interspecific hybrid ((*Triticum durum* × *Th. ponticum*) × Pyrotrix 28); WWH—wheat–wheatgrass hybrid. IT—infection type: 0—without symptoms; ;—small necrotic spots; 4—large pustules.

### 3.3. Interaction of P. graminis f. sp. tritici with Wheat Protected by the Sr24, Sr25, and Sr26 Genes

Five *Pgt* isolates from the population 2022 with typical ITs (0, ;, ;1, 2-) were used to study the effects of the *Sr24*, *Sr25*, and *Sr26* genes. Significant differences of the isolates in spore germination at the surface of susceptible and resistant wheat were not found (Table 3). All isolates formed significantly fewer appressoria on introgressive lines compared to the control. In seven out of nine combinations 'isolate—line', the appressoria's development rate was close to that noted on the *Th. ponticum* and WWHs (21.8–39.4%). The incompatibility of isolate 1 with the line Tc*Lr24/Sr24* and *Th. ponticum* were equal in the appressoria development, orientation of growing tubes, and penetration into the stomata (27%, 38%, and 3.8%, respectively). On all introgressive lines, three isolates (1, 2, and 4) showed a violation of the germ tubes' orientation to the stomata at the level of *Th. ponticum* or WWHs (38–67%). Large differences were noted between the combinations in penetration into the stomata, from low to moderate (3.5–25.6%). Two of these indicators were combined randomly when the isolates developed on the lines with the *Sr24*, *Sr25*, and *Sr26*. In two combinations with IT 2-, the isolates had the best indicators of appressoria on stomata and penetration into tissues (isolate 3—Tc*Lr24/Sr24* and isolate 5—Tc*Lr19/Sr25)*. In general, as a result of developmental disorders, the main part of the inoculum died at the surface of introgressive lines, and only a small proportion penetrated into the stomata (0.4–14.6%). The exceptions were combinations with IT 2-, in which the intensity of penetration into the stomata of lines with the *Sr24* and *Sr25* was higher (11.2–14.6%). A comparison of the isolate indicators on the lines showed that the suppression of the appressoria development was a common feature of the interactions. In terms of orientation to the stomata and penetration into them, the results differ and combine independently (Table 4).

In some combinations, the fungus was able to penetrate into the stomata and to develop single colonies in the tissues. In the cv. Chernyava 13, the mycelium grew fast, and 18–25 haustoria were formed in it (on average 21.5 haustoria per colony). In 5 days after inoculation, the colonies occupied an area of 24–46 thousand $\mu m^2$ (on average 27.8 thousand $\mu m^2$). The results of *Pgt* development in lines with *Sr24*, *Sr25*, and *Sr26* depended on ITs. The development of isolates in 20–50% of cases stopped at the SV stage in immune combinations (ITs 0, ;) (Figure 4a–c). Haustorium formation in colonies was significantly suppressed in comparison with the susceptible control. The analysis of the development of isolates Nos. 1, 2, 3, 4, and 5 in ILs with *Sr24*, *Sr25*, and *Sr26* showed that a colony's square (in total for all combinations) closely was correlated with haustorium number in them (r = 0.85) (Figure 4d). The smallest colonies with an area less than 2 thousand $\mu m^2$ had no haustoria (Figure 3g). In combinations with ITs ;1 and 2-, the colonies of

different sizes with a small number of haustoria was formed in the tissues. Colonies with 3–5 haustoria had an area of 4000–10,000 thousand $\mu m^2$. Small and medium-sized colonies stopped developing 3–5 days after inoculation, and their cells were vacuolized (Figure 3h). Near such colonies, the color of plant cytoplasm became more intense, which indicates an increase in their membrane permeability (Figure 3i). However, the typical HR, with characteristic collapse and rapid cytoplasm destruction, was not noted. In some cases, larger colonies were formed, but almost devoid of haustoria (Figure 3j). Over time, chlorotic and necrotic spots of different sizes appeared in their places. Colonies with 12–15 haustoria that occupied an area of at least 16 thousand $\mu m^2$ were able to develop small pustules. Thus, after *Pgt* penetration into the tissues, the incompatibility with lines manifested itself in the suppression of haustorial development without HR appearance. In most cases, the fungus died at the early stages, and only colonies with a larger haustorium number were able to develop the urediniopustules.

**Table 3.** Results of the development of the infection structures of *P. graminis* f. sp. *tritici* on wheat with *Sr24*, *Sr25*, and *Sr26* genes.

| Sample | Isolate | IT | Proportion of Germinated Spores, % | Proportion of Growing Tubes with Appressoria, % | Proportion of Appressoria, % | | Proportion of Inoculum Penetrated into Stomata, % |
|---|---|---|---|---|---|---|---|
| | | | | | On Stomata from the Total Number | Germinated from the Number on the Stomata | |
| Chernyava 13—control * | 1, 2, 3, 4 | 4 | 93.5 ± 3.2 | 85.0 ± 4.5 | 87.0 ± 3.6 | 88.0 ± 1.2 | 60.8 |
| TcLr24/Sr24 | 1 | 0 | 89.6 ± 2.4 | 27.6 ± 2.1 | 38.3 ± 2.9 | 3.8 ± 1.5 | 0.4 |
| | 2 | ;1 | 88.7 ± 3.5 | 39.4 ± 6.1 | 83.8 ± 4.9 | 5.4 ± 1.2 | 1.6 |
| | 3 | 2- | 92.2 ± 1.6 | 57.7 ± 4.5 | 91.4 ± 1.2 | 21.2 ± 3.5 | 11.2 |
| TcLr19/Sr25 | 1 | 0 | 93.8 ± 4.4 | 21.8 ± 2.3 | 86.3 ± 5.2 | 3.7 ± 1.3 | 0.7 |
| | 4 | ;1 | 87.6 ± 4.6 | 30.3 ± 2.9 | 45.0 ± 8.3 | 24.2 ± 3.2 | 3.3 |
| | 5 | 2- | 85.4 ± 2.4 | 65.3 ± 5.1 | 87.2 ± 3.2 | 25.6 ± 2.6 | 14.6 |
| LMPG-Sr26 | 2 | 0 | 89.8 ± 2.3 | 29.3 ± 3.5 | 66.9 ± 6.3 | 3.5 ± 1.1 | 0.7 |
| | 3 | ; | 93.4 ± 1.8 | 35.3 ± 5.1 | 88.5 ± 2.4 | 12.3 ± 2.3 | 3.8 |
| | 5 | ;1 | 93.3 ± 2.8 | 34.2 ± 2.3 | 88.1 ± 2.3 | 13.5 ± 1.4 | 3.9 |

Note: * average for 4 isolates; IT: 0—without symptoms; ;—small necrotic spots; ;1—medium necrotic spots and small pustules surrounded by medium necrotic zones; 1—small necrotic spots and small pustules with small necrotic zones; 2- —chlorotic spots and microscopic pustules surrounded by large chlorotic zones; 4—large pustules.

**Table 4.** Analysis of the similarity of the development indicators of *P. graminis* f. sp. *tritici* in interactions with introgressive lines.

| Isolate | Proportion of Growing Tubes with Appressoria | Proportion of Appressoria, % | |
|---|---|---|---|
| | | On Stomata from the Total Number | Germinated from the Number on the Stomata |
| | | *Sr24* and *Sr25* | |
| 1 | + | − | + |
| | | *Sr24* and *Sr26* | |
| 3 | ± | + | − |
| 2 | + | ± | + |
| | | *Sr25* and *Sr26* | |
| 5 | ± | + | ± |

Note: + similarity; ± moderate similarity; − significant differences.

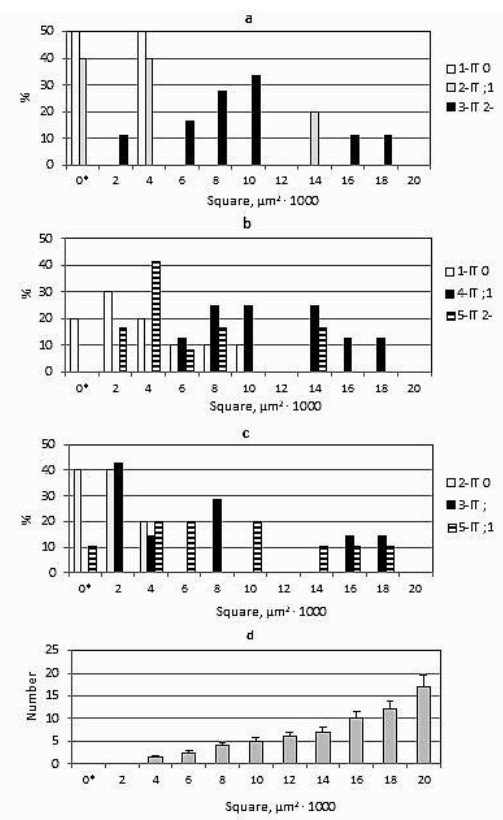

**Figure 4.** Distribution by square of colonies *P. graminis* f. sp. *tritici* in the tissues of the introgressive wheat lines Tc*Lr24/Sr24* (**a**), Tc*Lr19/Sr25* (**b**), and LMPG-*Sr26* (**c**), and the number of haustoria in the colonies (**d**). * substomal vesicle. 1, 2, 3, 4, and 5—the isolates. IT—infection type. IT: 0—without symptoms; ;—small necrotic spots; ;1—medium necrotic spots and small pustules surrounded by medium necrotic zones; 2- —chlorotic spots and microscopic pustules surrounded by large chlorotic zones.

### 3.4. The Role of ROS in Interactions of P. graminis f. sp. tritici with Th. ponticum and the Line with the Sr25 Gene

To understand the regulation of NHR and its possible mechanisms in introgressive lines, it is important to study the early stages of interaction with rust fungi. In our experiments, the role of $Ca^{2+}$ influx and ROS generation on the development of stem rust fungus was studied on plants that were previously treated with the verapamil (inhibitor of $Ca^{2+}$-channels) and SA in concentrations suitable to the induction of oxidative burst [76]. Visual estimation showed that the treatment of *Th. ponticum*'s plants with both reagents did not change the ITs (Table 5). Treatment with verapamil led to the suppression of the pustule development on the cv. Chernyava 13 and Tc*Lr19/Sr25* line, and only chlorotic spots of different sizes were noted on the leaves. The pretreatment of the cv. Chernyava 13 with SA led to a decrease in IT from 4 to 2+ and a reduction in the number of pustules. The resistant line Tc*Lr19/Sr25* showed only minor necrotic spots (IT ;) after SA treatment.

$O_2^{\bullet-}$ and $H_2O_2$ generation was determined using specific NBT and DAB dyes, respectively. In the case of pathogen development on the surface of the susceptible plants, NBT stained only granular structures in the fungal cells, which was related to the detection of dehydrogenase activity in mitochondria (Figure 5a). When infected of *Th. ponticum*, $O_2^{\bullet-}$ was detected in the places of the appressoria's contact with the guard cells, such as in the cytoplasm of the appressoria (Figure 5b). Two days after inoculation, hydrogen peroxide was detected in fungal cytoplasm and in adjacent plant cells (Figure 5c,d). Later, the cytoplasm of the appressoria and SVs was destroyed, as evidenced by their dark staining (Figure 3d).

**Table 5.** The influence of the suppression or induction of ROS generation on the development of *P. graminis* f. sp. *tritici* on the *Th. ponticum* and the line Tc*Lr19/Sr25*.

| Isolate | Variant | IT | Proportion of Growing Tubes with Appressoria, % | Proportion Appressoria, % | | Proportion of Structures Penetrated into the Stomata, % | | | | | | | | | |
|---|---|---|---|---|---|---|---|---|---|---|---|---|---|---|---|
| | | | | On Stomata from the Total Number | Germinated from the Number on the Stomata | SV | Colony Square, $\mu m^2$ | | | | | | | | |
| | | | | | | | 2001–4000 | 4001–6000 | 6001–8000 | 8001–10,000 | 10,001–12,000 | 14,001–16,000 | 16,001–20,000 | >25,000 |
| | | | | | | Chernyava 13—control | | | | | | | | |
| 1 | Control | 4 | 87.5 ± 4.6 | 90.1 ± 3.5 | 90.2 ± 3.6 | 0.0 | 0.0 | 0.0 | 0.0 | 0.0 | 0.0 | 0.0 | 0.0 | 100.0 |
| | Verapamil | ; | 84.6 ± 3.7 | 87.6 ± 4.4 | 87.0 ± 3.8 | 0.0 | 0.0 | 0.0 | 0.0 | 20.0 | 26.7 | 40.0 | 13.3 | 0.0 |
| | SA | 2+ | 86.3 ± 4.1 | 91.2 ± 3.7 | 88.6 ± 4.1 | 0.0 | 0.0 | 0.0 | 0.0 | 0,0 | 20.0 | 33.3 | 46.7 | 0.0 |
| | | | | | | *Th. ponticum* | | | | | | | | |
| 1 | Control | 0 | 14.1 ± 0.2 | 39.3 ± 3.5 | 0.0 | 0.0 | 0.0 | 0.0 | 0.0 | 0.0 | 0.0 | 0.0 | 0.0 | 0.0 |
| | Verapamil | 0 | 13.9 ± 0.2 | 40.2 ± 3.2 | 19.1 ± 1.3 | 100.0 | 0.0 | 0.0 | 0.0 | 0.0 | 0.0 | 0.0 | 0.0 | 0.0 |
| | SA | 0 | 14.3 ± 0.3 | 41.0 ± 4.2 | 0.0 | 0.0 | 0.0 | 0.0 | 0.0 | 0.0 | 0.0 | 0.0 | 0.0 | 0.0 |
| | | | | | | Tc*Lr19/Sr25* | | | | | | | | |
| 1 | Control | 0 | 21.8 ± 2.3 | 86.3 ± 5.2 | 3.7 ± 1.3 | 20.0 | 30.0 | 20.0 | 10.0 | 10.0 | 0.0 | 0.0 | 0.0 | 0.0 |
| | Verapamil | 0 | 24.1 ± 2.1 | 83.1 ± 3.8 | 10.2 ± 2.1 | 33.3 | 33.3 | 22.2 | 11.4 | 0.0 | 0.0 | 0.0 | 0.0 | 0.0 |
| | SA | 0 | 23.3 ± 3.2 | 85.0 ± 4.7 | 0.0 | 0.0 | 0.0 | 0.0 | 0.0 | 0.0 | 0.0 | 0.0 | 0.0 | 0.0 |
| 4 | Control | ;1 | 30.3 ± 0.9 | 45.0 ± 8.3 | 24.2 ± 3.2 | 0.0 | 0.0 | 0.0 | 13.2 | 25.3 | 25.5 | 25.0 | 11.2 | 0.0 |
| | Verapamil | ; | 32.5 ± 1.5 | 42.3 ± 5.2 | 30.1 ± 5.1 | 0.0 | 0.0 | 0.0 | 30.0 | 40.0 | 30.0 | 0.0 | 0.0 | 0.0 |
| | SA | ; | 33.1 ± 2.6 | 43.0 ± 3.9 | 7.2 ± 2.3 | 19.2 | 11.5 | 14.0 | 24.7 | 23.1 | 7.5 | 0.0 | 0.0 | 0.0 |
| 5 | Control | 2- | 29.3 ± 3.5 | 66.9 ± 6.3 | 25.6 ± 2.7 | 0.0 | 17.5 | 32.5 | 11.4 | 22.8 | 0.0 | 25.7 | 0.0 | 0.0 |
| | Verapamil | ; | 25.6 ± 3.1 | 63.1 ± 4.5 | 28.5 ± 3.5 | 0.0 | 22.0 | 35.0 | 28.0 | 15.0 | 0.0 | 0.0 | 0.0 | 0.0 |
| | SA | ; | 28.1 ± 2.9 | 68.5 ± 4.3 | 9.8 ± 1.8 | 21.4 | 21.4 | 21.4 | 28.6 | 7.14 | 0 | 0 | 0.0 | 0.0 |

Note: IT—infection type: 0—without symptoms; ;—small necrotic spots; ;1—medium necrotic spots and small pustules surrounded by medium necrotic zones; 2- —chlorotic spots and microscopic pustules surrounded by large chlorotic zones; 2+—medium pustules; 4—large pustules; SV—substomal vesicle.

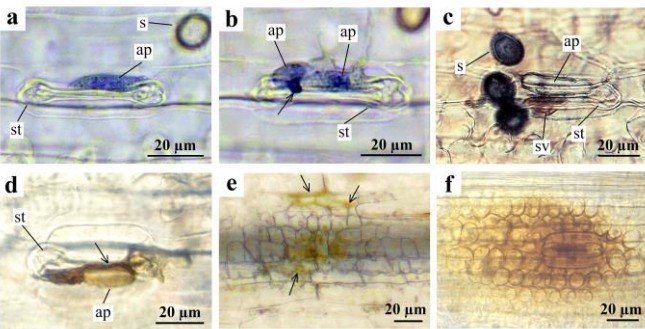

**Figure 5.** ROS generation in interactions of *P. graminis* f. sp. *tritici* with common wheat (**a**), *Th. ponticum* (**b**–**d**), and introgressive wheat line with *Sr25* (**e**,**f**): (**a**)—appressorium on the stoma of cv. Chernyava 13, with the stained granules being the mitochondria; (**b**)—superoxide anion in penetrating peg of appressorium (arrow); (**c**)—hydrogen peroxide in the cytoplasm of dead substomal vesicle; (**d**)—hydrogen peroxide in the cytoplasm of substomal vesicles located on the guard cell, and in plant walls contacted with the appressorium (arrow); (**e**)—slow accumulation of $H_2O_2$ on plant cell walls in the zone of the dying colony (arrows); (**f**)—increased accumulation of $H_2O_2$ in the zone of the colony in the tissue of the plant pretreated with SA. Staining: (**a**,**b**)—NBT; (**e**,**f**)—DAB.

When infected with an isolate with IT 0, the introgressive line with *Sr25*, ROS occurred similarly to that of *Th. ponticum*. When the line was infected with isolates with IT ;1 and 2-, a superoxide anion was detected less frequently at the points of contact with the appressoria, and penetration into the stomata was more successful (Table 5). In the zone of small- and medium-sized colonies, a slow amount of $H_2O_2$ accumulated on plant cell walls after the finishing of fungus development (Figure 5e). Probably, such ROS generation was a reaction to the products of pathogen damage (DAMPs).

The pretreatment of plants with verapamil did not affect the development of the fungus on the surface until being in contact with stomata. The inhibition of $Ca^{2+}$-channels in the cv. Chernyava 13 did not affect the penetration of the fungus into the stomata. However, the size of the colonies in the treated plants decreased significantly (Table 5). Colonies died in the late stages of development without HR (Figure 3l), and medium-size chlorotic spots formed instead of pustules. After the treatment of *Th. ponticum* with verapamyl, $O_2^{\bullet-}$ and $H_2O_2$ were not detected. The fungus penetration into non-host stomata significantly increased (up to 10–15%), but the development was limited to one infection hyphae (Figure 3k). The pretreatment of the tall wheatgrass with SA led to the enhanced accumulation of $O_2^{\bullet-}$ in the appressorium cytoplasm and their rapid death on the stomata (Table 5).

In the plants with *Sr25*, the treatment with verapamil led to a significant increase in the penetration of isolate 1 (IT 0) into the stomata, but there was no significant progress in the development of small colonies (Table 5). When the plants were inoculated with isolate 4 (IT ;1), the penetration into the stomata increased by 20%, and the area of colonies in the tissues increased to an average size (6–10 thousand $\mu m^2$), but development stopped before sporogenesis. The accumulation of $H_2O_2$ and HR in the zone of the colony has not been established. The penetration of isolate 5 (IT 2-) into the stomata decreased by only 10%, and the sizes of colonies did not change significantly.

The induction of ROS generation in plants with *Sr25* led to the accumulation of $O_2^{\bullet-}$ and the rapid death of all appressoria of the isolate 1 (IT 0) on the stomata. When studying the development of isolates 4 and 5 (ITs ;1 and 2- respectively), it was found that, after the accumulation of superoxide anion, the death rates of appressoria sharply increased, and the sum of colonies in the tissues diminished (Table 5). In plant cytoplasm in the mycelium zone, the accumulation of $H_2O_2$ increased, and their staining enhanced, which indicates the changing of membrane permeability (Figure 5f, 3 m). As a result, the average size of colonies decreased by 2.5–3.2 times.

Thus, with the development of *Pgt* on *Th. ponticum* and the introgressive line with *Sr25*, calcium fluxes and ROS generation had the main effects at the stage of penetration into tissues and ensured the pre-haustorial resistance of plants. The infection with a set of avirulent isolates revealed quantitative differences in the manifestation of reactions in wheat with the *Sr25* gene.

## 4. Discussion

There are myriads of individuals in the populations of rust fungi on large-scale wheat monocrops. In these populations, active evolutionary processes are aimed at overcoming plant resistance [21,23,77]. The spreading rust fungi by the aerogenic way promotes the migration of virulent forms to new territories [78]. The long-term experience of using cultivars bred by CIMMYT in various regions of the world has shown that 12 genes are the most valuable for protecting wheat from stem rust. Two genes were obtained from common wheat (*Sr2* and *Sr23*). The rest were transferred from related cereals: *Th. ponticum* (*Sr24* and *Sr25*), *Secale cereale* (*Sr31*, *Sr1RS*[Amigo], and *Sr50*), *Triticum* spp. (*Sr36*, *Sr45*, and *SrTmp*), and *Aegilops* spp. (*Sr33* and *Sr38*) [6]. In Australia, *Sr26* has remained effective for several decades [79]. The tall wheatgrass genes *Sr25*, *Sr26*, *Sr43*, and *Sr61* provide resistance to the Ug99 race [34,80]. Prior to the emergence of the Ug99 race, virulence to *Sr24* was rare in the world [81], but five races virulent to it were identified in the Ug99 group by 2006 [21,36]. In the large territory of Russia, *Sr24* remains effective [19]. The experience of crop production has shown that most effective genes can be overcome with large-scale monocropping protected by a single gene. Therefore, in Australia, after the prolonged cultivation of cultivars with *Sr24*, virulent clones appeared. A similar situation developed in the Volga region, where the *Lr19/Sr25* translocation has been used in wheat breeding since 1980s [49]. However, *Sr24* and *Sr25* in combination with other genes provide a high resistance to the disease [49,79]. With the sharp increase in the harmfulness of stem rust, the regular monitoring of resistance sources and the effectiveness of introgressive genes is necessary. Completed studies showed that *Th. ponticum* and WWHs were immune to the highly aggressive West Siberian *Pgt* population, even with the epidemic of 2015. In 2015, lines with the *Sr24*, *Sr25*, and *Sr26* genes were moderately susceptible; however, by 2021, their resistance had been restored.

To breed cultivars with a durable resistance, information about the NHR regulation is important. The general hypothesis of the PTI-ETI was originally formulated based on the studies of the interactions between the model species *Arabidopsis thaliana* and a set of pathogens. It was proposed that NHR and host resistance provide different genetic systems and separate signaling cascades [59,60]. Later, it was revealed that the PTI and ETI signaling pathways can overlap, especially in host-related species [62]. More recently, in some models, it was shown that, after contact with PAMPs and effectors, a number of overlapping activities are manifested, including $Ca^{2+}$ influx, ROS generation, and MAPK cascade signaling [63,64]. Some plant proteins may participate in the regulation of PTI and ETI in plant interactions with bacteria and fungi [82,83]. In this regard, a revised model of plant immunity was proposed, in which ETI is not a separate system, but a module for the amplification of reactions dependent on PTI [22,64]. It is of important to check the revised model in the rust fungi–cereals pathosystems.

The general PTI-ETI model was formulated for a set of pathogens (bacteria, viruses, and fungi). The information about the different localization of PRR and NLR receptors and their structure was an important basis for the separation of PTI and ETI [84]. However, the pathogens greatly differ in their biology and patterns of interaction with plants. Viruses enter plants with the help of vectors or through wounds and move through cell pores along the symplast [85]. Phytopathogenic bacteria have a special system for introducing effectors into cells—Type-III-secret-system effectors (T3SEs) [86,87]. Necrotrophic and hemibiotrophic fungi secrete toxic effectors, and then develop in the destroyed tissues [88]. In this regard, the localization of NLRs to these pathogens in the cytoplasm is obvious. At the same time, rust fungi penetrate into organs through stomata [89] and develop in

photosynthetic tissues. Upon penetration into the cell, the haustoria do not destroy the plasmalemma, but invaginate it. As a result of fungal action, the extracellular PRRs relocate inside the cell. The extrahaustorial membrane is formed around the haustoria, as a modified part of the plasmalemma [66,90]. Receptors encoded by cultivar resistance genes (NLRs) are located on the extrahaustorial membrane, and after the recognition of an avirulent fungus, HR develops [90]. Thus, PRRs and NLRs that recognize rust fungi can be located inside the cell, which creates the basis for the interaction of PTI-ETI systems.

In recent years, nine *Sr* genes have been cloned (viz. *Sr13, Sr21, Sr22, Sr33, Sr35, Sr45, Sr46, Sr50,* and *Sr60*). Eight of them had an NLR-type structure (Nucleotide-binding, Leucine-rich-repeat immune Receptors), and *Sr60* encodes a tandem kinase protein [34]. Among them, *Sr13* was identified in common wheat, but others were introgressed from related species: *T. monococcum* (*Sr21, Sr22, Sr35, Sr45,* and *Sr60*), *Ae. squarrosa* (*Sr33*), *Ae. tauschii* (*Sr46*), and *S. cereale* (*Sr50*) [30–33,91]. At the same time, *Sr35* determined pre-haustorial resistance [92], characteristic for NHR. This shows that the gene of an immune species with NHR effect can have an NLR structure.

The model "*Pgt—Th. ponticum* and wheat with the *Sr24, Sr25,* and *Sr26* genes" is of interest for studying the regulation of cereal NHR to biotrophic rust fungus and the effects of introgressed genes. For the first time, it was shown that the NHR of *Th. ponticum* provide an effective suppression of *Pgt* on the plant surface. The incompatibility of *Pgt* with tall wheatgrass was manifested in at least three stages: the orientation of growing tubes to the stomata, development of appressoria, and penetration into the stomata. Quantitative differences in the interactions in these three stages were noted between accessions and WWHs. Such results can be explained by the presence of several key NHR genes in *Th. ponticum* and their different distribution between the accessions and WWHs.

Monogenic cultivar resistance is traditionally considered as a host (ETI), although in recent decades, many alien genes have been introduced into wheat. When studying the interaction of *Pgt* with wheat lines protected with *Sr24, Sr25,* and *Sr26*, it was found that the main part of the inoculum died on the surface or during penetration into the stomata, as on *Th. ponticum*. The action of introgressed genes led to a strong suppression of the appressoria's development, which in most combinations was equal or similar to that in *Th. ponticum*. This indicates that genetic material that allows the implementation of important NHR mechanisms was introgressed into the wheat. Quantitative differences in the pathogen suppression degree show certain differences in gene action. Some differences in the interactions of *Pgt* isolates with lines were found in the orientation to the stomata and penetration into the tissues. It is likely that isolates adapted to varying degrees to the surface properties of the lines appear in the Omsk *Pgt* population.

When studying NHR, much attention is paid to active reactions, such as ROS generation, $Ca^{2+}$ influx, and the activation of MAPK cascades [65,93]. ROS perform multiple functions in immunity, including toxic effects on pathogens, messenger in signaling, destruction of cytoplasm during HR, and lignification of cell walls [94,95]. The superoxide anion $O_2^{\bullet-}$ is formed by NADP·H oxidase localized on the outer side of the cell membrane. The enzyme is activated by the $Ca^{2+}$ fluxes that occur after the recognition of the pathogen's metabolites by surface receptors. $Ca^{2+}$ influx is important for the induction of PRR- and NLR-mediated immunity and downstream signaling [65,93,96,97]. The inhibitors of $Ca^{2+}$-channels (verapamil and staurosporin) suppress the ion movement and prevent $O_2^{\bullet-}$ generation [96]. Two peaks of ROS generation have been established in resistant cultivars [64]. The first peak appears a few minutes after the recognition of the effectors and is related to the activity of the constitutive NADP·H oxidase. The superoxide anion is an extremely toxic substance that has an antimicrobial effect and can damage plant cells. The enzyme superoxide dismutase (SOD) converts $O_2^{\bullet-}$ to the less toxic hydrogen peroxide $H_2O_2$ [98]. The second peak is linked to the *de novo* synthesis of oxidative enzymes (peroxidases and oxalate oxidases) [95]. Phytohormones are involved in the implementation of responses against biotrophic and necrotrophic pathogens (SA and jasmonic acids,

respectively) [63,95]. The oxidative burst can be enhanced by the pretreatment of plants with SA or its analogues [76,99].

According to our results, in *Th. ponticum* and lines with the *Sr24*, *Sr25*, and *Sr26*, the first generation of the superoxide anion occurred concurrently, when the appressoria was in contact with the stomatal guard cells. $O_2^{\bullet-}$ accumulated in the cytoplasm of the appressoria or SV, followed by the rapid destruction of cells. This indicates that, in *Th. ponticum*, the PRRs determining *Pgt* are located on the surface of guard cells. A similar ROS generation was determined in the lines with wheatgrass genes. The defense reactions of *Th. ponticum* and wheat lines were probably mediated by the same plant receptors and led to the inactivation of the pathogen at the pre-haustorial stage. Probably, the PRRs encoded by *Sr24*, *Sr25*, and *Sr26* recognize different components of the fungus cell wall. The role of ROS generation in the interaction with Pgt was confirmed in plants treated with verapamil or SA. Later, hydrogen peroxide accumulated in the destroyed structures of the fungus and adjacent plant cells. Our results confirm the opinion that the induction of ROS generation in PTI and ETI occurs simultaneously [64,65,100]. This opinion was formed in the course of studying the interaction of *A. thaliana* with incompatible *Pseudomonas syringae* bacteria.

Previously, *Pgt* development was studied on non-host *S. cereale* and wheat with the *Sr31* gene, which suppressed the development of stem rust worldwide for a long time [101]. The disorders of *Pgt* development on the plant surface of rye and *Th. ponticum* were similar, but the rye additionally violated spore germination. On wheat cultivars with *Sr31*, appressoria development and penetration into the stomata were significantly suppressed, which is similar to the action of wheatgrass genes. During the interaction of *Pgt* with wheat protected by *Sr50* (*Sr31* allele), it was shown that the overcoming of resistance was linked to the loss of the fungal effector encoded by the *AvrSr50* gene, which elicited defense reactions [92]. This shows that the mechanisms of NHR of *Th. ponticum* and *S. cereale* and the effect of genes introgressed from them are similar.

Earlier, similar facts were mentioned when the relative species *P. triticina* interacted with non-hosts (*Zea mays*, *Panicum miliaceum*, and *Avena sativa*). It was shown that the appressoria need to be tightly attached to guard cells to successfully penetrate into the stomatal slit. After that, extracellular $O_2^{\bullet-}$ generation took place at the contact points [69]. The study of the *P. triticina* interactions with immune wheat lines protected with transloca-tion *Lr19/Sr25* and the *Lr38* gene (from *Th. intermedium*) revealed a similar extracellular generation of superoxide anions by guard cells as an answer to contacts with appresso-ria [102]. The signs of the destruction of fungal hyphae and plant cell walls supporting the ROS action was determined by light and electron microscopy of immune Tc*Lr19/Sr25* plants infected with *P. triticina* [103]. More compatible isolates of *P. triticina* did not provoke ROS generation in guard cells; as a result, the fungus overcame pre-haustorial resistance and formed colonies and pustules in the Tc*Lr19/Sr25* line [75]. Such results suggest that the tall wheatgrass genes *Lr19* and *Sr25* act similarly on the related rust fungi *P. triticina* and *Pgt*. The presence of receptors recognizing alien metabolites in the guard cells and their participation in signaling was confirmed by the closure in response to the treatment of fungal elicitors (e.g., chitin and chitosan) [104,105].

When studying the model species *A. thaliana*, it was found that the stomatal guard cells play an important role in protecting against the invasion of non-pathogenic *E. coli* bacteria. The phenomenon was called "stomatal immunity" [106]. Later, it was found that recognition of MAMPs by PRRs induced PTI. Signaling was mediated by messengers (ROS, nitric oxide, and Ca influx), regulators of innate immune response (MPKs), and plant hormones [107,108]. SA and abscisic acid hormones acted as positive regulators of stomatal immunity [109]. The NHR of *Th. ponticum* and *S. cereale* to the rust fungi *Pgt* and *P. triticina* shows similarities with stomatal immunity, with adjustments for the biology of rust fungi. ROS generation by the stomatal guard cells indicates that PRRs are located on them, and the "friend–foe" border passes here. The similarity of the manifestations of NHR to different groups of pathogens (rust fungi and bacteria) in cereals and *A. thaliana* indicates that plants

have common defense mechanisms against the invasion of non-pathogenic or unadapted microorganisms into the stomata.

Due to the great progress in molecular technologies, engineering crops becomes possible. The transfer of genes encoding PRRs between species is considered a promising direction of the work [110]. The revealed features of the *Sr24*, *Sr25*, and *Sr26* genes show that they may be of interest for such engineering. In some regions of the world, it has been shown that, after the loss of the effectiveness of individual *Sr24* and *Sr25* genes, the cultivars with combinations *Sr24+Sr31* and *Sr25+Sr31* showed high resistance [49,79]. As shown by this and a previous study [101], both of these genes provide stomatal immunity to *Pgt*. Obviously, these genes recognize different components of the *Pgt* cell wall, whose loss means a fitness penalty for the pathogen.

As discussed earlier, tall wheatgrass had a strong suppressive effect on fungus development before contact with stomata. This indicates a significant role of preformed constitutional barriers in the interactions. Previously, it was found that the parasitic fungi need to receive a complex of stimuli from plants for their development. The lack of or false stimuli from non-hosts lead to the inhibition of unadapted and heterologous fungi [55]. The surface hydrophobicity, the structural features of the surface, and the shape of the stomata are important for the stimulation of fungi [111]. The enzymes cutinase and esterases are involved in the attachment of rust growing tubes to the surface of plants. The products of wax and cutin splitting serve as signals for recognizing the host and triggering signaling cascades (MAPKs and cAMP). A complex of stimuli induces the morphogenesis of infection structures [112,113]. After penetration into tissues, the formation of rust HMC and haustoria is induced by host substances, including the components of cell walls, as well as gaseous leaf secretions [114–116].

Introgressive lines with *Th. ponticum* genes suppressed *Pgt* both at the plant surface and in the tissues, that is, they had a pleiotropic effect on interactions. There was also a correlation between the strong suppression of appressoria development and the orientation of growing tubes and the early cessation of the colony growth in the tissues. The sizes of colonies closely correlated with the number of haustoria developed in plant cells. To fully develop and finish the life cycle, the rust fungi need to establish biotrophic relations with plants [90,115]. Through the haustoria, rust fungi receive nutrition in the form of hexoses, amino acids, minerals, and vitamins from plant cells [115]. The colonies with a smaller number of haustoria had vacuolized cells and were aborted soon after inoculation without HR. Probably, these colonies died from starvation. A weak accumulation of hydrogen peroxide was observed in plant cells after the death of the pathogen. This was probably due to the identification of fungal DAMPs by NLRs. $H_2O_2$ did not accumulate in verapamil-treated plants, but a decrease in colony size and the suppression of pustule development were found. Presumably, the suppression of fungus development after treatment with verapamil is associated with a violation of Ca admission, which is necessary for the metabolism of plants and pathogens. After SA treatment, the accumulation of $H_2O_2$ increased, with a simultaneous decrease in colony growth and sporogenesis. The negative effect of SA on mycelium growth is obviously linked to the development of systemic acquired resistance (SAR), since SA and its analogues induce SAR to biotrophic pathogens [117].

In general, the effects of *Sr24*, *Sr25*, and *Sr26* genes introgressed from wheatgrass lead to suppression of *Pgt* development on the surface and in plant tissues. This indicates that complex translocations with the *Lr24/Sr24* and *Lr19/Sr25* genes, as well as the alien fragment with the *Sr26* gene, have a pleiotropic effect on the pathogenesis. It is quite likely that, in the introgressed fragments, in addition to the genes encoding PRRs, there are genes that define effective preformed constitutional barriers. Additional research is needed to confirm this assumption.

## 5. Conclusions

*Th. ponticum* is a valuable source of resistance genes to the fungal diseases of wheat. The history of crop production has shown that the *Sr24*, *Sr25*, and *Sr26* genes transferred from *Th. ponticum*, individually or in combination, provide the cultivars with a high resistance to rust diseases in different regions of the world. Despite the active use of the genetic material of *Th. ponticum* in breeding, the mechanisms of NHR and the effects of the *Sr* genes introgressed to wheat have not yet been studied. In Western Siberia, a strong epidemic of stem rust occurred in 2015, and in the following years, the disease appeared on wheat crops regularly. *Th. ponticum* and WWHs created on its basis have shown immunity to the disease. Samples protected by the *Sr24*, *Sr25*, and *Sr26* genes showed moderate susceptibility in 2015, but by 2021, their resistance had recovered. It was found that NHR ensured *Pgt* death on the surface of *Th. ponticum* and WWHs due to the disruption of stomatal search, the formation of appressoria, and the penetration into the stomata. The *Sr24*, *Sr25*, and *Sr26* genes acted similarly in introgressive lines. In rare cases when penetration into tissues occurred, most colonies died due to the suppression of the haustorium formation from starvation without HR. The death of appressoria on the stomata of *Th. ponticum* and introgressive lines was linked to the generation of superoxide anions by the guard cells. Probably, ROS generation was induced by the recognition of the appressoria's cell wall components by PRR receptors. This phenomenon is similar to the stomatal immunity revealed by the interaction of *A. thaliana* with non-pathogenic bacteria.

The results show that the *Sr24*, *Sr25*, and *Sr26* genes allow for the appearance of important NHR mechanisms in wheat. This information is interesting for breeding cultivars with durable resistance to rust diseases and engineering crops using molecular technologies.

**Author Contributions:** L.P.: conceptualization, methodology, and writing and editing; V.P. and V.K.: field and laboratory investigations, visualization, and data analysis; V.P.: project administration. All authors have read and agreed to the published version of the manuscript.

**Funding:** The investigation was supported by the Russian Science Foundation (project No. 22-24-20067).

**Institutional Review Board Statement:** Not applicable.

**Informed Consent Statement:** Not applicable.

**Data Availability Statement:** Not applicable.

**Conflicts of Interest:** The authors declare no conflict of interest. The funders had no role in the design of the study; in the collection, analyses, or interpretation of data; in the writing of the manuscript, or in the decision to publish the results.

## Abbreviation
### Abbreviations in the text

| | |
|---|---|
| cv. | cultivar |
| DAMPs | Damage-Associated Molecular Patterns |
| ETI | Effector-Triggered Immunity |
| HMC | haustorial mother cell |
| HR | hypersensitive reaction |
| IT | infection type |
| *Lr* | leaf rust resistance gene |
| MAMPs | Microbe-Associated Molecular Patterns |
| NHR | nonhost resistance |
| NIL | near-isogenic line |
| NLRs | Nucleotide-binding, Leucine-rich-repeat immune Receptors |
| *Pgt* | *Puccinia graminis* f. sp. *tritici* |
| PTI | PAMP-triggered immunity |
| ROS | reactive oxygen species |
| PAMPs | Pathogen-Associated Molecular Patterns |

| PRRs | Pattern Recognition Receptors |
|---|---|
| SA | salicylic acid |
| *Sr* | stem rust resistance gene |
| SV | substomal vesicle |
| Tc | cultivar Thatcher |
| WWH | wheat–wheatgrass hybrid |
| *Yr* | stripe rust resistance gene |

**Abbreviations in Figures 1, 3, and 5**

| ap | appressorium |
|---|---|
| gt | growing tube |
| ha | haustoria |
| hmc | haustorial mother cell |
| HR | hypersensitive reaction |
| ih | infection hypha |
| s | urediniospore |
| st | stoma |
| sv | substomal vesicle |
| up | urediniopustule |

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
