# Peer review of "Nonhost Resistance of Thinopyrum ponticum to Puccinia graminis f. sp. tritici and the Effects of the Sr24, Sr25, and Sr26 Genes Introgressed to Wheat"

_2037-0164, doi:10.3390/ijpb14020034_

Round 1

Reviewer 1 Report

Dear sir, the paper 'Nonhost resistance of Thinopyrum ponticum to Puccinia graminis f. sp. tritici and effects of the Sr24, Sr25, and Sr26 genes introgressed to wheat' deserves publication in IJPB. The content is clear, and so the message. In general is well written, and the motivation of the authors can be perceived throughout the manuscript. I attach a file with minor mistakes I have seen along the ms. For instance, authors must replace the word 'violation' by 'incompatibility'.

Best

Author Response

Dear reviewer!

Thank you for your careful analysis of our manuscripts and useful comments.

Your comments and suggestions have been taken into account and the text was corrected.

Best regards!

Authors

Reviewer 2 Report

1. the introduction part ban be shortened. the plant immunity and inreaction between fungi and host, for example.

2. 3.1. Estimation of Stem Rust Development in the Field Conditions and Laboratory , Pictures for rust on stems of R and S   should be provided .

3. data analysis should be done for Table 2,and 3--.

4. Fig.2 and 4, should be improved.

Author Response

Dear reviewer!

Thank you for a thorough analysis of the manuscript and useful comments.

  • About shortening of the Introduction

In the introduction, we tried to justify the relevance of our experimental model. Th. ponticum and other species with genomes different from the common wheat are actively used to protect cultivars from diseases. Alien genes of these species often provide durable resistance of common wheat to rust diseases.

The other reviewers agree with our version of the Introduction. In this regard, we consider it possible to leave the Introduction unchanged.

  • The photo of the WWH-6 with IT 0 was added to Figure 2 (c).

From our point of view, it is more objective to give the pictures of seedlings on whose leaves the resistance mechanisms have been studied.

The results of the field evaluation of samples for 7 years are shown in Table 1. During the observation period, the evaluation results varied. Therefore, it is not entirely clear which photos should be provided.

  • The description of the Tables 2 and 3 was made in more details.
  • Figures 2 and 3 have been corrected

Best regards!

Authors

Reviewer 3 Report

Stem rust (causal agent Puccinia graminis f. sp. tritici) is one of the most destructive diseases of wheat, capable of completely destroying the yield of a susceptible cultivar during the epiphytotic development of the pathogen. In breeding programs for wheat resistance to biotic and abiotic factors, including resistance to stem rust, introgressive resistance genes from alien species such as Thinopyrum ponticum are widely used. The study of the mechanisms of nonhost resistance (NHR) is of undoubted fundamental and practical interest.

Thus the work of authors is definitely relevant and undoubtedly important.

The article is a high-level work and the authors obtained significant very interesting results. The article is very interesting, original, the content of the article corresponds to the abstract and title. The tables and figures are complementing the text well.

There are some comments and suggestions for authors.

1. You write that the inoculation was carried out using monopustular isolates of the fungus. Have fungal isolates been characterized by virulence genes, were the races of the fungus identified? The same applies to the P. graminis populations used in the work. If possible, please, add this information.

2. Please correct the Figure 2 (numbers on the y-axis overlap each other).

Other comments are presented in the text of the article.

 The authors have undoubtedly obtained important results, and I believe that the article can be published after revision.

Author Response

Dear reviewer!

Thank you for your careful analysis of our manuscripts and useful comments.

Your comments and suggestions have been taken into account and the text was corrected.

  • We consider it possible to leave the phrase you have highlighted in the Abstracts, because it reflects the new information received by us and shows the relationship with the results of leading laboratories.
  • The results of the isolate phenotyping were included into the Methods.

We do not study the Pgt population. Population monitoring is conducted by specialists of the Institute of Cytology and Genetics (Novosibirsk, Russia) (Refs. 19, 20).

Our results do not contradict with their data about race composition of the Omsk Pgt population

Best regards!

Authors
